# Lower DNA methylation levels in CpG island shores of *CR1*, *CLU*, and *PICALM* in the blood of Japanese Alzheimer's disease patients

Risa Mitsumori[1], Kazuya Sakaguchi[2¤a], Daichi Shigemizu[1], Taiki Mori[1¤b], Shintaro Akiyama[1], Kouichi Ozaki[1], Shumpei Niida[1], Nobuyoshi Shimoda[2]*

**1** Medical Genome Center, National Center for Geriatrics and Gerontology, Obu, Aichi, Japan, **2** Department of Regenerative Medicine, National Center for Geriatrics and Gerontology, Obu, Aichi, Japan

¤a Current address: Axcelead Drug Discovery Partners, Inc., Fujisawa, Kanagwa, Japan
¤b Current address: Department of Oral Microbiology, Asahi University School of Dentistry, Mizuho, Gifu, Japan
* shimoda@ncgg.go.jp

**Data Availability Statement:** All relevant data are within the paper and its Supporting Information files.

## Abstract

The aim of the present study was to (1) investigate the relationship between late-onset Alzheimer's disease (AD) and DNA methylation levels in six of the top seven AD-associated genes identified through a meta-analysis of recent genome wide association studies, *APOE*, *BIN1*, *PICALM*, *CR1*, *CLU*, and *ABCA7*, in blood, and (2) examine its applicability to the diagnosis of AD. We examined methylation differences at CpG island shores in the six genes using Sanger sequencing, and one of two groups of 48 AD patients and 48 elderly controls was used for a test or replication analysis. We found that methylation levels in three out of the six genes, *CR1*, *CLU*, and *PICALM*, were significantly lower in AD subjects. The combination of *CLU* methylation levels and the *APOE* genotype classified AD patients with AUC = 0.84 and 0.80 in the test and replication analyses, respectively. Our study implicates methylation differences at the CpG island shores of AD-associated genes in the onset of AD and suggests their diagnostic value.

## Introduction

Moderate concordance rates for Alzheimer's disease (AD) among genetically identical twins suggest the possible involvement of epigenetics in the etiology of AD [1, 2]. Among epigenetic components, DNA methylation and chromatin modifications are of great interest in AD because they have been reported to change with aging [3, 4], which is the main cause of the disease, and their unintended changes affect gene expression [5]. As DNA methylation differences in genomes between AD patients and cognitive normal elderly individuals are expected to not only reveal AD-susceptible genes, but also provide biomarkers for clinical purposes, they have been extensively examined in blood and the brain using candidate genes and genome-wide approaches [6–10]. Sanger or pyrosequencing and DNA methylation arrays have generally been employed for region-specific and genome-wide analyses, respectively.

**Funding:** This work was supported by The Research Funding for Longevity Sciences (21-16 and 29-20 to NS and 29-45 to KO) from National Center for Geriatrics and Gerontology and by the Japanese Ministry of Health, Labor, and Welfare (KO). K.S. (Kazuya Sakaguchi) is affiliated with Axcelead Drug Discovery Partners, Inc., Fujisawa, Kanagawa, Japan. The funder provided support in the form of salaries for K.S., but did not have any additional role in the study design, data collection and analysis, decision to publish, or preparation of the manuscript. The specific roles of the author are articulated in the 'author contributions' section.

**Competing interests:** The authors have declared that no competing interests exist. K.S. is affiliated with Axcelead Drug Discovery Partners, Inc., Fujisawa, Kanagawa, Japan. This does not alter our adherence to PLOS ONE policies on sharing data and materials.

Although large numbers of differentially methylated genes have been reported, only a few genes in the brain and blood have been independently confirmed [7, 11]. This suggests that methylation anomalies in AD samples are so marginal that depending on the sample size or sensitivity of the procedures employed, they cannot reproducibly be detected [12, 13]. Another possibility is that genomic regions with evident differences were not targeted by these studies. Whole-genome bisulfite sequencing of large samples of cognitively normal and AD subjects with sufficient coverage may be the only approach to circumvent these limitations; however, this is prohibitive for large-scale studies due to the associated costs.

We are interested in identifying differentially methylated regions (DMRs) in blood DNA that may be applied for the diagnosis of AD and its prognosis. We assumed that DMRs in AD-associated genes, such as those identified by genome-wide association studies (GWASs) [14, 15], in the brain or any other tissues confer susceptibility to the onset of AD through subtle, but long-lasting expression changes in these genes, similar to single nucleotide polymorphisms (SNPs). Based on this assumption, we examined whether DMRs exist in AD-associated genes in the blood of late-onset AD (LOAD) patients. We focused on CpG island shores in AD-associated genes because methylation levels at these regions are vulnerable under several conditions, such as tissue differentiation, reprogramming, aging, and disease, including AD [16–20]. CpG island shores are defined as 2-kb-long regions that lie on both sides of a CpG island [16]. CpG islands are, on average, 1000 base pairs (bp) in length, characterized by dense clusters of CpG dinucleotides, and located around the promoter regions of 70% of human genes [21]. CpG islands are typically exempt from DNA methylation irrespective of gene expression [22, 23], and the further away CpG dinucleotides are from CpG islands, the higher the chance of methylation [18, 24, 25]. Therefore, methylated cytosines are symmetrically distributed with respect to promoter CpG islands, and the cytosines of CpGs in CpG island shores are often slightly and moderately methylated [16]. A previous study demonstrated that gene expression levels were negatively associated with methylation levels at CpG island shores [16].

We demonstrated that, in blood, AD-related DMRs exist in CpG island shores in three out of six AD-associated genes examined, *CR1* (complement receptor 1), *CLU* (clusterin), and *PICALM* (phosphatidylinositol-binding clathrin assembly protein), and that with the combination of *APOE* (apolipoprotein E) genotypes, the methylation levels of *CLU* in blood may effectively differentiate AD patients from cognitive elderly with a high sensitivity and specificity.

## Materials and methods

### Ethics statements

The present study was conducted after receiving informed consent from all individuals and approval from the ethics committee of the National Center for Geriatrics and Gerontology (NCGG). The design and performance of the present study involving human subjects were clearly described in a research proposal. All participation was voluntary and informed consent was received in writing before registering at the NCGG biobank, which collects human biomaterials and data for geriatrics research. In instances in which the dementia specialist determined that the patient lacked capacity to consent due to substantial cognitive impairment, written informed consent was received from the legal next of kin with patient assent, which was previously approved by the local ethics committee.

### Patient sample recruitment

The genomic DNA of 293 subjects were extracted from peripheral blood leukocytes by standard protocols using Maxwell RSC Instrument (Maxwell RSC Buffy Coat DNA Kit, Promega,

USA) and the corresponding clinical data were provided by the NCGG Biobank. Ninety-six subjects were patients with AD, 40 with VaD, 34 with DLB, and 27 with FTD, and 96 subjects were cognitively normal elderly controls (hereafter, controls). One of two groups of 48 AD patients and 48 age- and gender-matched controls was used as a test group and the other as a replication group. The cognitive status and severity of dementia were assessed by the Mini-Mental State Examination (MMSE). The status of the *APOE e4* allele genotype (the main genetic risk factor for AD) and the MMSE score were obtained from the NCGG biobank. All subjects were >60 years in age. All control subjects had a MMSE score of >25.

## Methylation analyses

The methylation levels of all target regions except *TREM2* were first assessed by Sanger sequencing, and pyrosequencing was employed only for validation of the results. This was because the former method can survey longer regions (~450 bp), hence more CpGs, than the latter (~50 bp). In either method, 200 ng of genomic DNA was initially treated with sodium bisulfite to convert non-methylated cytosines to uracil using the EZ DNA Methylation-Gold Kit (Zymo research) following the manufacturer's instructions. Target regions were PCR amplified by primer sets designed by the web tool, MethPrimer [26], for Sanger sequencing or by Pyrosequencing Assay Design Software ver.2.0 provided by Qiagen for pyrosequencing. The conditions of PCR, lengths of amplicons, and sequences of primers are shown in S1 Table. All PCR amplicons were run on 2% agarose gels to confirm successful amplification, as shown in S12 Fig. For Sanger sequencing, amplicons were cloned into the pGEM-T vector (Promega), which was then used to transform the *E. coli* strain, DH5α (Takara Bio). Plasmid DNA was amplified from each of 32 colonies by Templiphi (GE Healthcare) and used as a template for a dideoxy sequencing reaction (BigDye ver.3.1, Applied Biosystems). Sequences were elucidated on a Genetic Analyzer 3500 or 3130xL (Applied Biosystems), and data containing a minimum of twenty sequences were analyzed by QUMA [27] to quantify the mean methylation levels of PCR products. For pyrosequencing, the mean methylation levels of PCR amplicons were directly quantified on PyroMark Q48 (Qiagen). PyroMark Q48 Autoprep Software provides the percent methylation of each CpG site being sequenced along with the quality scores of blue, yellow, or red, which is determined by the intensity of the light signal (Relative Light Unit; RLU) that is released upon incorporation of a nucleotide. A methylation value that receives a blue score indicates that it was measured with a high RLU of ≥20 and is thus high-quality data. A yellow score indicates intermediate quality with an RLU of ≥10. Red scores indicate low quality and suggest failed pyrosequencing analysis. We employed pyrosequencing results composed of only blue, or blue and yellow scores (S7 Fig). PCR amplicons were pre-pared again for the samples from which sequencing results containing red scores were obtained.

## Identification of CpG island shores

To identify CpG island shores in AD-associated genes, we plotted CpG dinucleotide sites along the genes using the designated program CyGnusPlotter, which was written for the present study and is freely available via the internet (https://github.com/kzyskgch/CyGnusPlotter). These diagrams revealed that, as reported in many human genes, there was a cluster of CpGs at the promoter regions, termed CpG island, which was free of DNA methylation. As CpG island shores are defined as regions that immediately flank CpG islands, the exact boundary was only identified after the edges of the hypomethylated region were confirmed. We located approximate positions of the edges based on markedly high CpG density in CpG islands [16] and CyGnusPlotter visualized the difference in the density. We thus selected regions close to

CpG island but with a notably lower CpG density for candidates of CpG island shores to be tested.

## SNP genotyping and RNA-sequencing data

The three SNPs associated with the risk of AD on *CR1*, *CLU*, and *PICALM* chromosome loci (rs3818361, rs11136000 and rs3851179) were genotyped in 148 NCGG samples, representing 80 patients with AD and 68 cognitively normal (CN) adults. We further performed RNA-sequencing analysis of 32 blood samples, composed of 5 ADs and 27 CNs [28]. The gene expression data were normalized using the trimmed mean of M values (TMM) [29].

## Statistical analyses

Comparisons between two and more groups were made using the Student's *t*-test and ANOVA, respectively, by PRISM ver.5 (GraphPad Software, Inc., San Diego, CA, USA). The sensitivity and specificity of the measured variables for the diagnosis of AD were evaluated using a receiver operating characteristic (ROC) curve. ROC plots were drawn by Minitab 17 (Minitab Inc.). Regarding multiple biomarkers, logistic regression analyses were conducted to derive analytical expression for the risk of AD using methylation levels as continuous variables and the *APOE* genotype as a nominal variable. The classification performance of *CLU* methylation, the *APOE* genotype, and the combination of both was assessed using the area under the ROC curve (AUC). The ROC curve is a plot of the probability of correctly classifying positive samples against the rate of incorrectly classifying true negative samples. Therefore, the AUC measure of an ROC plot is a measure of predictive accuracy. A DeLong test was used to compare the AUC between groups [30]. All tests were two-tailed and a *p* value <0.05 was considered to be significant.

## Results

### Selection of target regions

We selected target regions in the CpG island shores of six of the top seven AD-associated genes in a large scale genome-wide association meta-analysis of clinically diagnosed late-onset AD [31] for methylation analyses, *APOE*, *CLU*, *CR1*, *PICALM*, *BIN1*, and *ABCA7*, as follows: we initially referred to the distribution of CpG dinucleotides along the genes (Fig 1 and S1–S5 Figs). Promoter CpG islands were easily recognized as a cluster of CpG dinucleotides around the 5' ends of the genes, and regions juxtaposed to the promoter CpG islands were putative CpG island shores. We PCR-amplified portions of the putative CpG island shores at which at least several CpGs existed and quantified their methylation levels. Slightly (~10%) or moderately (~50%) methylated portions that met the criteria for CpG island shores were used in subsequent analyses. These portions were only found in a limited space in the case of *APOE* (region II in Fig 1).

### DNA methylation differences

We quantified the methylation levels of target regions in the six genes in 48 control and AD blood samples by cloning and sequencing PCR products from bisulfite-treated genomic DNAs. Demographic data for the samples included in the present study are shown in Table 1. When stratified by the disease status, we found that the mean methylation levels in three out of the six genes, *PICALM*, *CR1*, and *CLU*, were significantly lower in AD than in control samples, whereas no significant differences were detected in the other three genes, *APOE*, *BIN1*, or *ABCA7* (Fig 2). The presence of methylation differences in half of the six genes suggested that

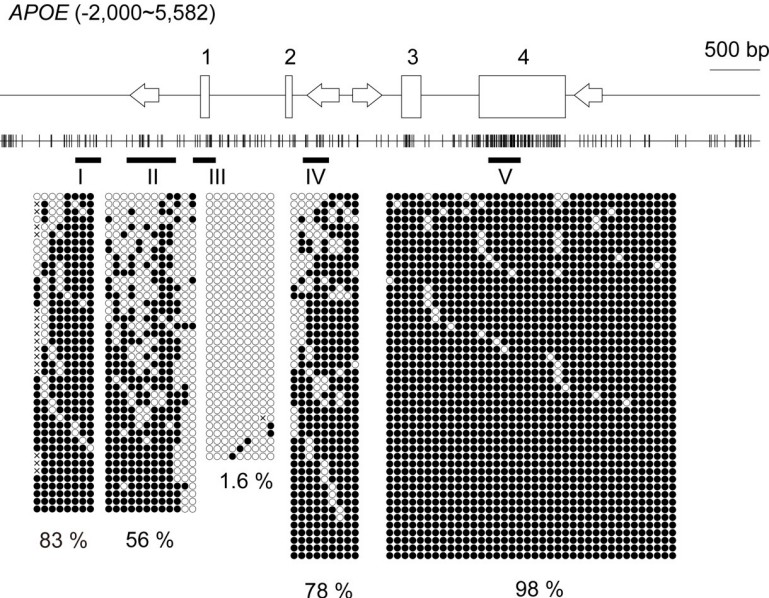

**Fig 1. Relative positions of CpG dinucleotides selected for the methylation analysis in *APOE*.** The schematic shows the distribution of CpG dinucleotides along *APOE*. The position of the transcription start site is defined as +1; hence, "-2,000" in the parenthesis indicates that the diagram includes 2 kb upstream of *APOE*, whereas "5,582" in the parenthesis is the position 2 kb downstream from the end of the last exon. Vertical lines indicate the positions of CpG dinucleotides. Open rectangles depict exons and exon numbers are shown above the rectangles. Four arrows indicate *Alu* elements in the locus and their orientations relative to the coding strand of *APOE* [32]. The fundamentals of these diagrams were automatically drawn by the web-based tool CyGnusPlotter. It collects the genomic structure of the most representative isoform of a requested gene from the Ensembl database with upstream and downstream regions of the designated lengths. Black horizontal bars with Greek letters (I-V) indicate the regions for bisulfite sequencing. Filled and open circles represent methylated and unmethylated cytosines, respectively. Each column represents a unique CpG site in the examined amplicon and each line is an individual DNA clone. Crosses indicate positions at which CpG is absent due to DNA polymorphisms. The percentages of the mean methylation levels derived from four subjects are shown at the bottom. There was a conventional CpG island (region III) that covers exon 1. Moderate (~50%) DNA methylation was observed at a limited region (II) juxtaposed to the CpG island, a region called the "CpG island shore".

the development of AD was related to methylation changes in the CpG island shores of AD-associated genes. AD-related hypomethylation in blood samples was also reported for *TREM2* (Triggering receptor expressed on myeloid cells 2), another AD-associated gene [33]. We

**Table 1. Characteristics of subjects.**

|  | Test (n = 96) | | Replication (n = 96) | | Non-AD dementia (n = 95) | | |
|---|---|---|---|---|---|---|---|
|  | Controls | AD | Controls | AD | DLB | VaD[a] | FTLD[b] |
|  | (n = 48) | (n = 48) | (n = 48) | (n = 48) | (n = 34) | (n = 40) | (n = 27) |
| Age, mean (SD) | 71.9 (3.7) | 72.7 (5.4) | 70.2 (1.5) | 70.5 (1.8) | 79 (5.4) | 79 (6.9) | 70.2 (8.8) |
| Female, No (%) | 22 (46) | 24 (50) | 26 (54) | 24 (50) | 21 (62) | 15 (38) | 17 (63) |
| MMSE score, mean(SD) | 28.5 (1.5) | 13.3 (5.1) | 29.4 (0.8) | 17.7 (3.8) | 16.7 (4.7) | 17.4 (4.8) | 16.6 (6.9) |
| *APOE ε4* homozygote, No. (%) | 1 (2.1) | 9 (19) | 0 (0) | 4 (8.3) | 3 (8.8) | 1 (2.6) | 0 (0) |
| *APOE ε4* heterozygote, No. (%) | 9 (19) | 22 (46) | 6 (13) | 28 (58) | 9 (26) | 7 (18) | 10 (37) |
| *APOE ε4* non-carriers, No. (%) | 38 (79) | 17 (35) | 42 (88) | 16 (33) | 22 (65) | 31 (79) | 17 (63) |

[a]The MMSE score of one subject and the *APOE* genotype of another subject were not available.

[b]The MMSE score of one subject was not available. Abbreviations: *APOE ε4*, the *ε4* allele of *APOE*; ROC, receiver operating characteristic; AD, Alzheimer's disease; DLB, dementia with Lewy bodies; VaD, vascular dementia; FTD, frontotemporal dementia.

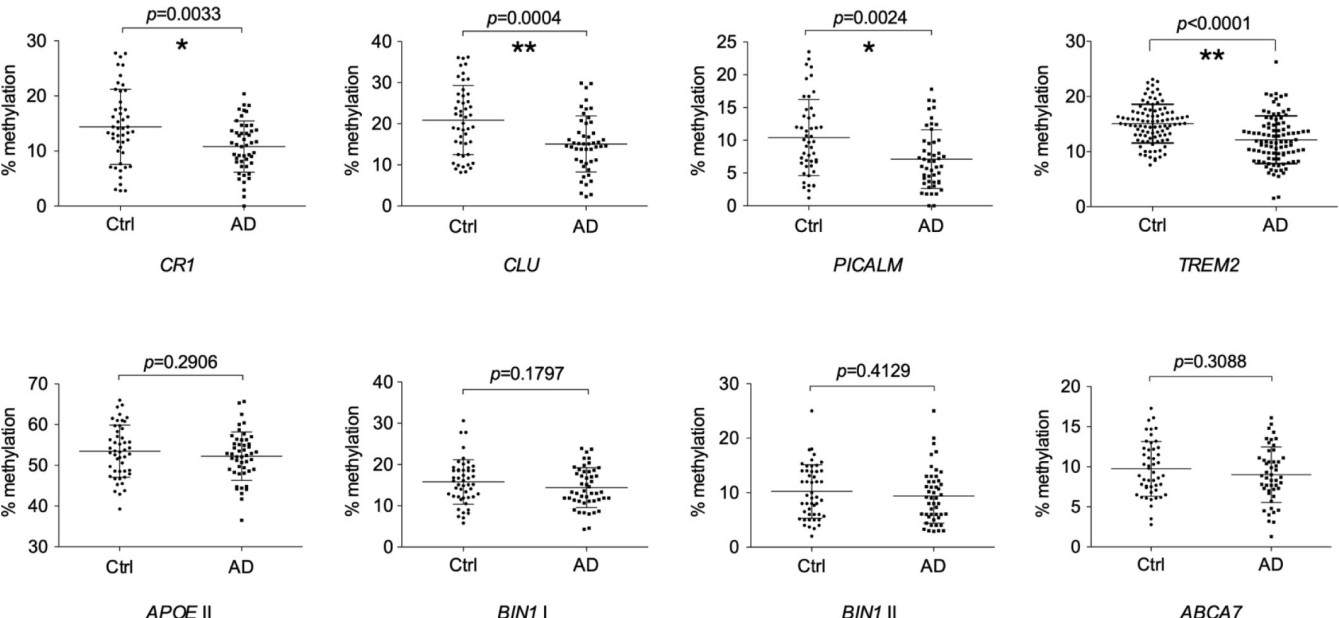

**Fig 2. Methylation rates at eight CpG island shores in seven AD-associated genes in the peripheral blood of control and AD subjects.** Each dot represents the mean methylation rate at the CpG island shores of the indicated genes in a subject. Subjects were composed of a group of 48 cognitive normal elderly (Ctrl) and 48 AD patients, named the test group, except for *TREM2*, for which another set of 96 subjects, named the replication group, was also examined. Methylation rates were quantified by Sanger sequencing, except for *TREM2*, for which pyrosequencing was employed [33]. The relative position of the CpG island shore investigated in *APOE* was region II in Fig 1, whereas those in the other six genes are shown in S1–S6 Figs. The horizontal lines and error bars indicate the mean ± standard deviation. *p*-values were obtained using the Student's *t*-test. Statistical significance was defined at $^*p < 0.00625$, $^{**}p < 0.00125$ following Bonferroni correction. Abbreviations: Ctrl, control; AD, Alzheimer's disease.

judged that the region surveyed in their study corresponded to the CpG island shore of *TREM2* based on its location (S6 Fig) and methylation level [33]. We compared the methylation levels of the same region in our control and AD samples by following the procedure reported previously. As shown in Fig 2, the mean methylation level was significantly lower in AD than in control samples, which was consistent with previous findings [33].

## Replication and verification of methylation differences

To assess the reproducibility of the results obtained above, we compared the methylation levels of *APOE* and *CLU* in the replication set of 48 blood samples of control and AD using the same method as in the test set, and confirmed the absence of methylation difference in *APOE* and *CLU*, respectively (Fig 3A and 3B). We then confirmed the methylation levels of *CLU* in two sets of blood samples using pyrosequencing, which directly measures the methylation levels of PCR products (Fig 3C and 3D and S7 Fig). Lower methylation levels in AD samples were again confirmed in either set of blood samples. This suggests that methylation differences between controls and AD patients were reproducible.

## Correlation between gene expression and methylation levels

To investigate the correlation between gene expression and methylation levels, we performed linear regression analysis for *CR1*, *CLU*, and *PICALM* using TMMs from RNA-seq data and average methylation rates in the CpG island shores of 32 samples. However, there were no significant associations between gene expression and methylation levels in the three genes (S8 Fig).

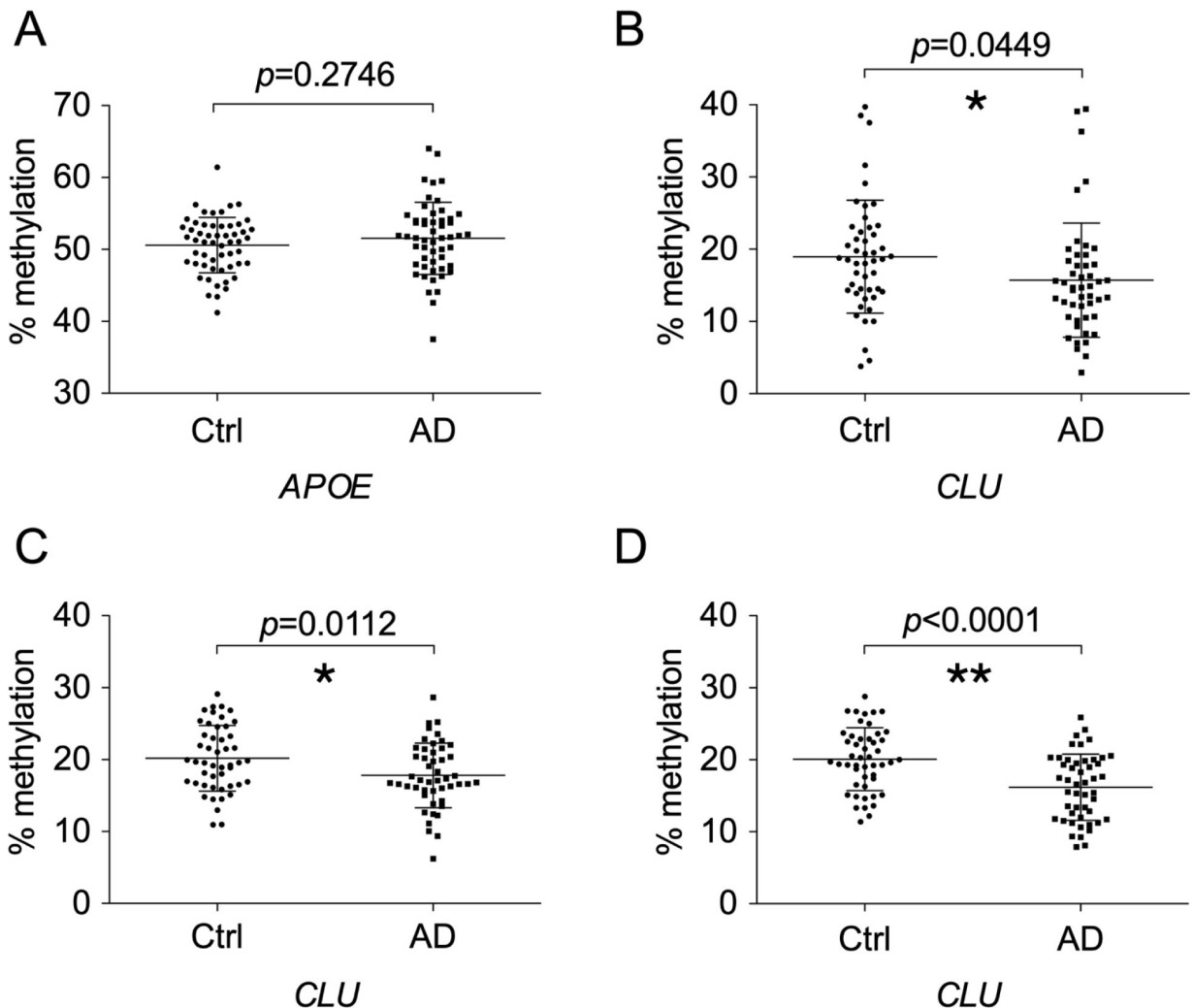

**Fig 3. Confirmation of the absence and presence of methylation differences at CpG island shores of *APOE* and *CLU*, respectively, between control and AD subjects.** Methylation rates in the replication group were quantified by Sanger sequencing for the CpG island shores of *APOE* (A) and *CLU* (B). Pyrosequencing was also employed for *CLU* to examine methylation differences in the test (C) and replication groups (D). The positions surveyed by Sanger sequencing were the same as those in Fig 2. The position of the genomic region in *CLU* quantified by pyrosequencing is shown in S2 Fig. Abbreviations: Ctrl, control subjects; AD, Alzheimer's disease subjects.

### Effects of SNPs, MMSE, and age on methylation

SNPs that correlate with the methylation degree of nearby CpG sites in cis are called methylation quantitative trait loci (mQTLs). Previous studies suggested that DNA methylation acts as an intermediary of genetic risk [34–36] and mQTLs associated with AD have been reported [37, 38]. As the six AD-associated genes examined above were all discovered through GWAS, AD-associated SNPs exist that are linked to *CLU*, *CR1*, and *PICALM*, in which lower methylated regions in AD blood were identified. Therefore, these SNPs may cause methylation differences in the three genes. To examine this possibility, we assessed the relationship between AD-associated SNPs and the methylation degree, but found no relationship in any of the genes (Fig 4A–4C). This demonstrated that these SNPs and methylation are independently associated with the onset of AD; therefore, the SNPs are not mQTLs. The independence of AD pathology-associated methylation changes from AD risk variants has also been reported [10, 36].

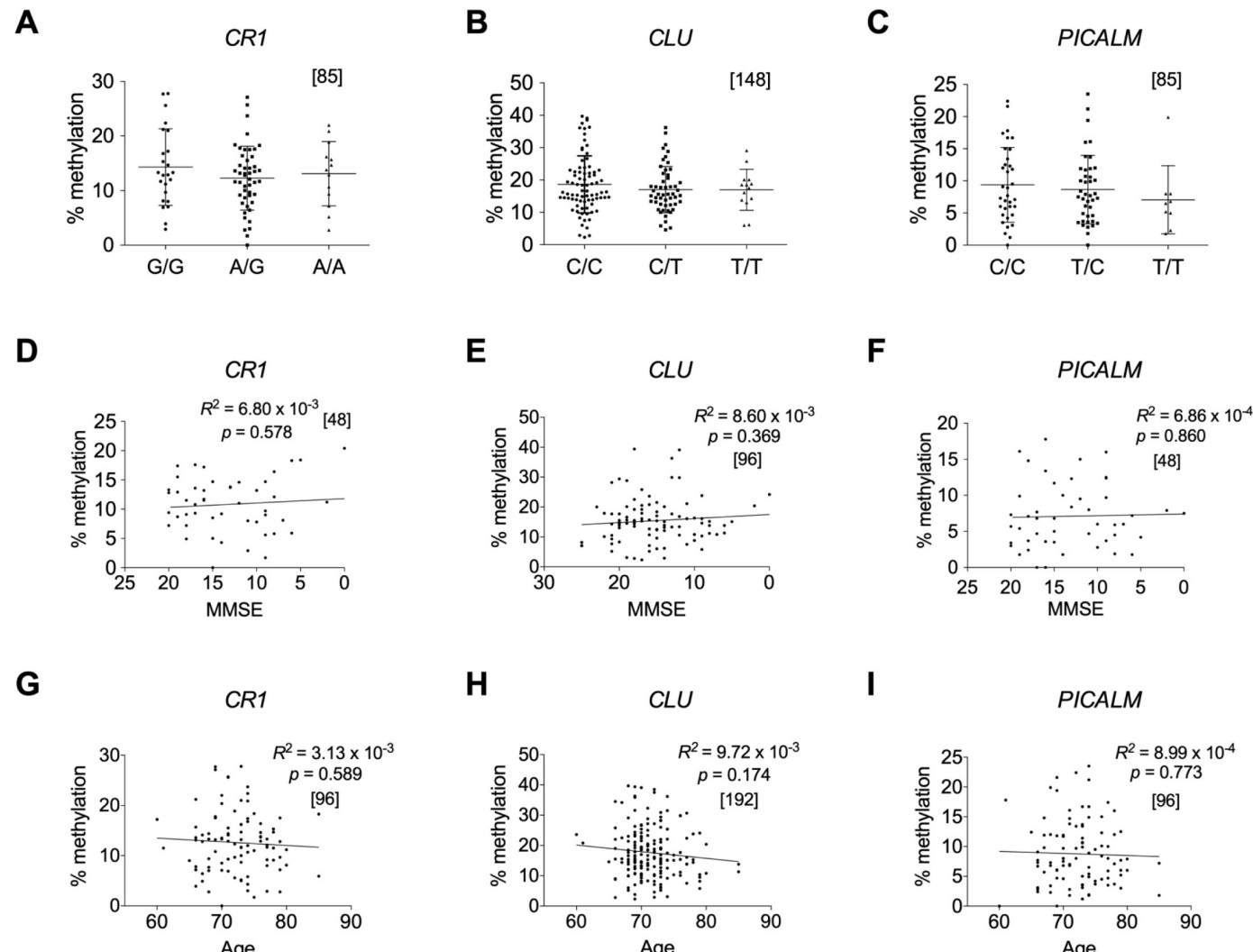

**Fig 4. Methylation rates in AD-associated DMRs are independent of known AD-associated SNPs, MMSE, and age.** Methylation rates in AD-associated DMR in *CR1*, *CLU*, and *PICALM* were plotted against the genotypes of AD-associated SNPs in *CR1* (rs3818361) (A) and *CLU* (rs11136000), (B) and upstream of *PICALM* (rs3851179) (C). No significant differences were detected in any combinations of the genotypes in any regions by the Kruskal-Wallis test [non-parametric one-way ANOVA] with Dunn's multiple comparison post-test. Methylation rates in DMR in *CR1*, *CLU*, and *PICALM* are also plotted against MMSE scores (D, E, F) and age (G, H, I), respectively. Relationships were analyzed based on Pearson's correlation coefficient. The numbers of samples examined are shown in parentheses. Methylation rates were derived from AD and control subjects in the test group for (A), (C), (G), and (I), and from those in the test and replication groups for (B) and (H). Data from AD subjects in the test group or in the test and replication groups were used to plot methylation rates against MMSE for (D) and (F) or for €, respectively. All methylation rates were quantified at the same regions as in Fig 2 by Sanger sequencing. Relative positions of SNPs in the three genes and information on the SNPs genotyped in the test and replication groups are shown in S1–S3 Figs and S2 Table, respectively. Abbreviations: Ctrl subjects, control; AD, Alzheimer's disease subjects; DMRs, differentially methylated regions; SNPs, single nucleotide polymorphisms; MMSE, Mini-Mental State Examination; ANOVA, analysis of variance.

We, however, found that *APOE ε4*, the highest risk allele of AD, may be a mQTL because a higher methylation level was detected in the *ε4* hyomozygotes (S9 Fig).

We investigated whether the degree of methylation in the three genes was related to the scores of the Mini-Mental State Examination (MMSE) and chronological age (Fig 4D–4I) based on Pearson's correlation coefficient. However, no relationships were observed in any of the three genes. Thus, methylation levels in the genes do not reflect the severity of disease and they are unrelated to aging in the Japanese elderly.

## Methylation in non-AD dementia

To clarify how specific the methylation changes detected are in dementia, we examined the methylation levels of *APOE* and *CLU* in blood samples of control, AD, dementia with Lewy bodies (DLB), vascular dementia (VaD), and frontotemporal dementia (FTD). Although no significant differences were observed in the methylation levels of *APOE* between the control and any dementia (Fig 5A), *CLU* methylation levels were lower in DLB than in the control, similar to AD samples (Fig 5B). This suggests that *CLU* hypomethylation is related to limited types of dementia.

## Blood-brain methylation discordance

In view of etiology, it is important to establish whether methylation changes in blood also occur in the brains of the same individuals. As we had no brain samples available from the same individuals that provided the blood samples used in this study, we needed to clarify whether the degree of DNA methylation in the brain reflects that in blood in the same individuals at the three AD-associated hypomethylated regions. To achieve this, we utilized two online searchable databases, designated BECon (Blood-Brain Epigenetic Concordance) [39] and the Blood Brain DNA methylation Comparison Tool [40], both of which describe the extent to which DNA methylation at a given CpG site in blood is related to that in three or four different regions, respectively, in the brains of matched individuals (S10 and S11 Figs). Both studies employed the Illumina 450K HumanMethylation array (HM450K), which has five probes against the regions we surveyed: three probes for the *CLU* region, and one each for *CR1* and *PICALM*. We found that the CpG sites targeted by the probes in the brain exhibited either low or moderate methylation levels, suggesting that they were in the CpG island shores. However, interindividual variations in the methylation levels of these CpGs were lower in the brain than in blood, and no correlation was noted between blood and any brain regions, except for a CpG site in *PICALM* in the entorhinal cortex (S10C Fig). Based on these results, overall methylation levels in the CpG island shores of the three genes in blood do not reflect those in brain.

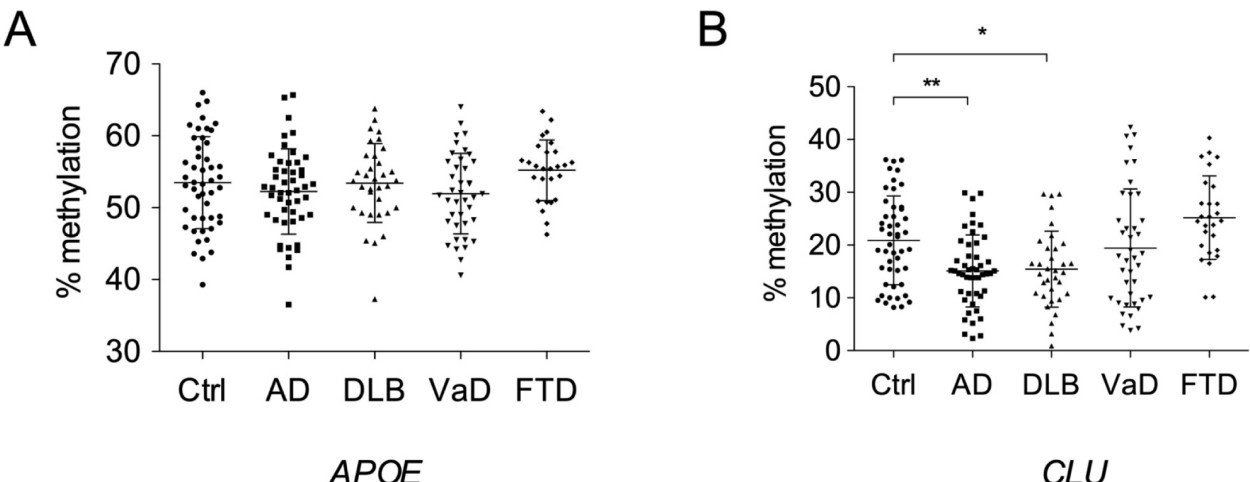

**Fig 5. Methylation rates at CpG island shores of *APOE* and *CLU* in non-AD dementia.** Methylation rates at the CpG island shores of *APOE* and *CLU* in DLB, VaD, and FTD were quantified by Sanger sequencing, and differences relative to control subjects in the test group were assessed using one-way ANOVA with Dunnett's multiple comparison post-test. The positions of the CpG island shores examined were the same as those in Fig 2. Abbreviations: Ctrl, control; AD, Alzheimer's disease; DLB, dementia with Lewy bodies; VaD, vascular dementia; FTD, frontotemporal dementia.

## Classification utility

The identification of subjects at high risk of developing AD is important for early interventions and clinical trials. As the methylation difference alone in either *CR1*, *CLU*, or *PICALM* was not sufficient for diagnosis, multiple logistic regression (MLR) analyses were used to derive linear classifier models that differentiate control and AD subjects using the data of test and replication groups (S4 Table). Methylation levels of *CLU* in the test and replication groups were employed for the analyses with the number of *APOE* ε4 alleles as covariates because the *APOE* epsilon4 (ε4) allele is the strongest genetic risk factor for AD. The performance of each model was assessed by the ROC curve, which establishes classification values, and AUC, which assesses multi-marker classification performance. Stepwise feature selection selected the top-performing linear combination of *CLU* methylation and the *APOE* genotype as MLR variables in both groups, which yielded AUCs of 0.83 and 0.85 using the data of the test and replication groups, respectively (Fig 6A). As slightly better classification performance was obtained using the data of the replication group, we applied the model to the data of the test group, which still yielded a high AUC of 0.80 (Fig 6B). In both cases, the differences in AUCs between *APOE* ε4 data alone and the combination of methylation and *APOE* ε4 data were significant (DeLong's test; *p* = 0.0083 for A, *p* = 0.0064 for B).

## Discussion

In the present study, Sanger sequencing of more than twenty thousand clones from 96 blood samples quantified the methylation levels of the CpG island shores of six of the top seven AD-associated genes in the latest meta-analysis [31]. Although it was low-throughput, we detected decreases in DNA methylation in three out of the six genes, *CLU*, *CR1*, and *PICALM*, whereas no significant DNA methylation changes were observed in *APOE*, *BIN1*, and *ABCA7*. Furthermore, we replicated AD-associated DNA hypomethylation in the CpG island shore of *TREM2* in our blood samples using pyrosequencing. Regarding *TREM2*, AD-related hypomethylation is not limited to a population in Japan because the donors who were recruited for the study that first reported *TREM2* hypomethylation in AD blood and those recruited for this study were from western and middle areas in Japan, respectively [33]. However, it may still be Japanese specific, and confirmation of methylation differences in these genes in other ethnic populations is needed to use them for clinical markers as described below.

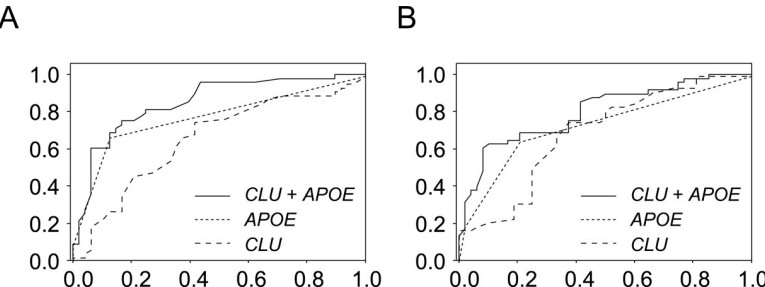

**Fig 6. ROC results for classifying control and AD subjects.** ROC analyses using either methylation rates at the CpG island shore of *CLU* (long dashed line), the number of the *APOE* ε4 allele (short dashed line), or the combination of both (solid line). A plot of ROC results from the replication group (A). The AUCs obtained were 0.66 for the methylation rate only (95% CI: 0.55–0.77), 0.78 for the *APOE* ε4 allele number only (95% CI: 0.68–0.87), and 0.85 for both (95% CI: 0.77–0.93). The model developed from the replication group was applied to the test group to draw another ROC plot (B). The AUCs obtained were 0.69 for the methylation rate only (95% CI: 0.56–0.80), 0.73 for the *APOE* ε4 allele number only (95% CI: 0.63–0.83), and 0.80 for both (95% CI: 0.71–0.89). ROC plots represent sensitivity (i.e., true positive rate) versus 1-specificity (i.e., false positive rate). Abbreviations: ROC, receiver operating characteristic; AD, Alzheimer's disease; AUC; area under the curve; CI, confidence interval.

Absence of methylation differences in the same region in *ABCA7*, as we surveyed, or upstream of a possible AD-associated gene, *TOMMO40*, was reported by another group [41, 42]. It is therefore unlikely that all AD-associated genes identified by GWAS carry a methylation difference. However, the regions surveyed in their- and our studies were limited in these genes; therefore, we cannot exclude the possibility of AD-associated methylation changes in other regions in these genes. Alternatively, AD-associated DNA methylation changes may develop in a tissue-specific manner because CpG dinucleotides with AD pathology-related methylation were found in the *ABCA7* and *BIN1* loci in the brain DNA [10, 20, 43].

We found that lower methylation levels in *CLU*, *CR1*, and *PICALM* were not associated with AD-associated SNPs in or around the genes, suggesting that methylation differences are an independent risk factor for AD onset. Contrary to these observations, we found an AD-related mQTL in *APOE*: a higher methylation level at the CpG island shore of *APOE* when the risk allele, $\varepsilon4$, was homozygous (S9 Fig), which suggests that *APOE* $\varepsilon4$ confers susceptibility to AD synergistically with methylation at the CpG island shore. However, two previous studies did not detect methylation difference in three CpGs positioned immediately upstream of the CpG island shore [44, 45], and considering the small methylation difference we found in *APOE* $\varepsilon4$ homozygotes, the possible synergistic effect needs to be ascertained in larger samples, preferably from other populations.

Genome-wide surveys that aimed at detecting AD-related methylation changes in blood have been conducted using Illumina HM450K and 850K (MethylationEPIC) arrays; however, AD-associated hypomethylation was not detected in *CR1*, *CLU*, or *PICALM* [8, 46, 47]. This discrepancy may be explained by the modest mean differences detected in *CR1*, *CLU*, and *PICALM*, which were all less than 10%, because depending on the probes, these arrays are not sufficiently sensitive to detect small differences, especially when sample numbers are limited [12, 13]. Alternatively, multiple testing correction employed in those analyses may have been too conservative and led to type II errors because no significant methylation differences were observed in any single CpG even in the three genes after Bonferroni correction ($p < 0.88 \times 10^{-3}$) (S2 Table).

It is reasonable to assume that methylation changes relevant to AD, if any, occur in the brain, a prominently affected tissue, and that blood methylation changes most likely reflect peripheral responses to the disorder rather than causally related variations. Thus, methylation changes in blood have been searched for as surrogate markers for brain methylation changes in most cases. Contrary to expectations, co-methylation changes that occur in both the blood and brains of AD patients were shown to be limited to a subset of DNA methylation sites [8, 46]. Methylation changes in the brains with neuropathology were not replicated in CD4+ lymphocytes [48] or in the whole blood in AD [9, 49] from the same individuals. Similarly, *in silico* analyses in the present study demonstrated no relationships between methylation in blood and the brain at CpG sites in AD-associated DMRs, and the DMRs found in blood were not detected in previous studies that searched for DMRs in AD brains [9, 10]. These results were consistent with the majority of DNA methylation sites, where interindividual variations in whole blood are not a strong predictor for those in the brain [40]. Therefore, the methylation changes detected in blood in the present study may not occur in AD brains. On the other hand, the chance of AD-associated methylation changes in three out of the six AD-linked genes in blood was more than coincidence. A previous study reported that age-related cognitive impairment was improved by the introduction of young blood in old mice (heterochronic parabiosis) [50, 51]; therefore, epigenetic changes that confer susceptibility to AD are not necessarily confined to neuronal cells and may also occur in blood.

A possible change that concomitantly occurs with the hypomethylation of CpG island shores is an increase in the expression of genes because CpG island shore methylation is

strongly and negatively related to gene expression [16]. Indeed, the inverse correlation is true for *TREM2* in blood [33] and it may be the case for *CLU* because the transcript levels of *CLU* were high in the blood of AD patients [52, 53]. To test this possibility, we performed linear regression analyses between *CR1*, *CLU*, and *PICALM* expression levels and average methylation rates in the CpG island shores of these genes, but no significant correlation was observed. As the number of the subjects was small (n = 32), which may lead to a Type II error, we should evaluate the correlation with higher numbers of samples in a future study. Based on the long duration of this disease, even small changes in DNA methylation in AD-associated genes, such as those observed in the present study, that accompany small changes in their expression may confer susceptibility to the progression of AD.

Although further studies are needed to clarify whether hypomethylation in AD-associated genes in blood is causal or consequential, these DNA methylation signatures have potential as clinical biomarkers for AD. We demonstrated that in combination with the *APOE* genotype, *CLU* methylation in CpG island shores provides good predictive performance for the diagnosis of AD in our subjects. As the methylation levels of *CR1*, *CLU*, and *PICALM* are not related to MMSE scores, methylation levels in the CpG island shores of these genes in the blood of AD patients may have already been lower than those in controls before the onset of the disease. This is worth testing using samples in longitudinal studies. Furthermore, we discovered that the hypomethylation of *CLU* occurred in the blood of DLB, but not in VaD or FTD. This suggests that hypomethylation is a disease-specific phenomenon and that it may reflect shared underlying pathophysiological mechanisms between the two diseases, as suggested in a previous study [54]. On the other hand, these results may be explained, in part, by possible diagnostic misclassification or the comorbidity of DLB with AD. Neurodegenerative disease-specific differential DNA methylation has also been reported for *ANK1* [55].

Ideal biomarkers for AD were proposed to have a sensitivity and specificity of >80% [56]; however, the diagnostic ability obtained by the combinatorial use of *CLU* methylation and the *APOE* genotype did not meet these criteria. Therefore, better DMRs for the clinical use of blood DNA in AD need to be identified. Further efforts to identify DMRs in other AD-associated genes will be a rational approach of choice. The success of combined uses of methylation rates and another variable regarding AD, such as the neuritic plaque burden [9, 10] and cognitive measure, [8, 57], may be referred to for unbiased screening of AD-associated methylation markers in blood.

## Supporting information

**S1 Fig. Relative positions of CpG dinucleotides selected for methylation analysis of *CR1*.** The schematic shows the distribution of CpG dinucleotides along *CR1*. The position of the transcription start site is defined as +1; hence, "-2,000" in the parenthesis indicates that the diagram includes 2 kb upstream of *CR1*, whereas "146,501" in the parenthesis is the position 2 kb downstream from the end of the last exon. Vertical lines indicate the positions of CpG dinucleotides. Open rectangles depict exons and a region spanning the first exon is enlarged. The fundamentals of these diagrams were automatically drawn by the web-based tool CyGnusPlotter. It collects the genomic structure of the most representative isoform of a requested gene from the Ensembl database with upstream and downstream regions of the designated lengths. Underlined is the region for which the methylation level was analyzed by bisulfite sequencing. It included a CpG to which Illumina designed a probe for the 450k array and gave the ID "cg14726637". The position of the AD-associated SNP, rs3818361, is also shown. The thick horizontal line represents the position of amplicons for bisulfite sequencing. (PDF)

**S2 Fig. Relative positions of CpG dinucleotides selected for the methylation analysis of *CLU*.** The figure is drawn similar to that in S1 Fig, except for the dashed line, which indicates the region in which methylation levels were quantified by pyrosequencing.
(PDF)

**S3 Fig. Relative positions of CpG dinucleotides selected for the methylation analysis of *PICALM*.** The figure is drawn similar to that in S1 Fig. The AD-associated risk SNP, rs3851179, which is linked to the gene, is not shown in the figure because it is located ~80 kb upstream.
(PDF)

**S4 Fig. Relative positions of CpG dinucleotides selected for methylation analysis of *ABCA7*.** The figure is drawn similar to that in S1 Fig.
(PDF)

**S5 Fig. Relative positions of CpG dinucleotides selected for methylation analysis of *BIN1*.** The figure is drawn similar to that in S1 Fig. In this gene, there are two regions (I and II) for which methylation levels were quantified by bisulfite sequencing.
(PDF)

**S6 Fig. Relative positions of CpG dinucleotides selected for the methylation analysis of *TREM2*.** The figure is drawn similar to that in S1 Fig except that the underlined region is the region for which methylation levels were analyzed by pyrosequencing.
(PDF)

**S7 Fig. Examples of Pyrograms$^{TM}$ that visualize methylation levels of CpG sites in a target sequence.** The plots shown are Pyrograms$^{TM}$, automatically generated by PyroMark Q48 Autoprep Software after pyrosequencing. The short sequences in the Pyrogram$^{TM}$ show the bisulfite-converted target sequences in which cytosines to be analyzed for methylation are indicated as Y (= C/T). The positions of Ys correspond to grey rectangles in the Pyrogram$^{TM}$. The numbers in small blue and yellow boxes indicate % methylation with high and intermediate sequencing qualities, respectively. The long rectangle in orange covers a cytosine that is not followed by guanine but by adenine in the template sequence of *TREM2*, which should therefore be converted to thymine by bisulfite treatment and PCR. Full bisulfite conversion was confirmed by no peaks at the cytosine position.
(PPTX)

**S8 Fig. Correlation between gene expression and average CpG methylation rates at CpG island shores.** Methylation rates in AD-related DMRs in *CR1*, *CLU*, and *PICALM* were plotted against the expression levels of *CR1*, *CLU*, and *PICALM*. Relationships were analyzed based on Pearson's correlation coefficient. The numbers of samples examined are shown in parentheses. All methylation rates were quantified at the same regions as in Fig 2 by Sanger sequencing.
(EPS)

**S9 Fig. Correlation between *APOE ε4* zygosity and *APOE* methylation.** Subjects were stratified by the zygosity of the *APOE ε4* allele and methylation rates at the CpG island shore of *APOE* were quantified by Sanger sequencing. The position of the CpG island shore examined was the same as that in Fig 2. The numbers of samples of each zygosity are shown in parentheses. Differences were assessed using one-way ANOVA with Tukey's multiple comparison post-test.
(EPS)

**S10 Fig.** *In silico* **examination of blood-brain methylation concordance by the online tool BECon.** Each plot shows the inter-individual variability of the methylation level at a CpG site across 16 subjects. The five CpGs examined were derived from three AD-associated DMRs identified in the present study: one CpG in *CR1* (A), three CpG sites in *CLU* (B), and one CpG site in *PICALM* (C). The numbers in "Correlation" columns in the tables indicate Spearman's rank correlation coefficients ($r_S$ or $\rho$), which were obtained by comparisons of methylation levels between blood and either one of three different cortical regions (Broadmann area 10 (BA10), prefrontal cortex; Broadmann area 7 (BA7), parietal cortex; and Broadmann area 20 (BA20), temporal cortex).
(PDF)

**S11 Fig.** *In silico* **examination of blood-brain methylation concordance by the Blood Brain DNA Methylation Comparison Tool.** Methylation levels at the five CpGs in S7 Fig, a CpG site in *CR1* (A), three CpG sites in *CLU* (B), and a CpG site in *PICALM* (C) in blood and four brain regions (PFC, prefrontal cortex; EC, entorhinal cortex; STG, superior temporal gyrus; CER, cerebellum) from the same individual donors were plotted in the rectangles, and the correlation of DNA methylation in blood with the four brain regions are plotted in square boxes. Methylation data were generated by Hannon et al. [40].
(PDF)

**S12 Fig. Verification of PCR amplification.** The indicated regions were PCR amplified from the bisulfite-treated blood DNA of two subjects with designated primer sets shown in S1 Table, and the amplicons were run on a 2% agarose gel. M: 100-bp DNA ladder (New England Biolabs).
(EPS)

**S1 Table. Primers and PCR conditions for bisulfite genomic sequencing.**
(DOCX)

**S2 Table. Statistical analyses of the differences of methylation at single CpG sites between control and AD.**
(XLSX)

**S3 Table. Association of known AD-associated SNPs with AD in NCGG samples (test and replication groups).**
(DOC)

**S4 Table. Results of binominal regression analyses.**
(XLSX)

## Acknowledgments

We thank the NCGG Biobank for providing the study materials, clinical information, and technical support.

## Author Contributions

**Conceptualization:** Nobuyoshi Shimoda.

**Formal analysis:** Risa Mitsumori, Kazuya Sakaguchi, Daichi Shigemizu, Taiki Mori, Shintaro Akiyama, Kouichi Ozaki, Nobuyoshi Shimoda.

**Funding acquisition:** Nobuyoshi Shimoda.

**Resources:** Shumpei Niida.

**Software:** Kazuya Sakaguchi.

**Supervision:** Nobuyoshi Shimoda.

**Writing – original draft:** Nobuyoshi Shimoda.

**Writing – review & editing:** Daichi Shigemizu, Taiki Mori, Shintaro Akiyama, Kouichi Ozaki.

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
