## [Decision Letter · Decision Letter 0]

1 Apr 2020

PONE-D-20-06667

Lower DNA methylation levels in CpG island shores of CR1, CLU, and PICALM genes in the blood of Alzheimer’s disease patients

PLOS ONE

Dear Dr. Shimoda,

Thank you for submitting your manuscript to PLOS ONE. After careful consideration by 2 Reviewers and an Academic Editor, the Reviewers had a discrepant view of the submission. Accordingly, all of the critiques of both Reviewers, especially Reviewer #1, must be addressed in detail in a revision to determine publication status. If you are prepared to undertake the work required, I would be pleased to reconsider my decision, but revision of the original submission without directly addressing the critiques of the two Reviewers does not guarantee acceptance for publication in PLOS ONE. If the authors do not feel that the queries can be addressed, please consider submitting to another publication medium. A revised submission will be sent out for re-review. The authors are urged to have the manuscript given a hard copyedit for syntax and grammar.

**Comments to the Author**

1. Is the manuscript technically sound, and do the data support the conclusions?

Reviewer #1: Partly

Reviewer #2: Yes

2. Has the statistical analysis been performed appropriately and rigorously? 

Reviewer #1: No

Reviewer #2: Yes

3. Have the authors made all data underlying the findings in their manuscript fully available?

Reviewer #1: Yes

Reviewer #2: Yes

4. Is the manuscript presented in an intelligible fashion and written in standard English?

Reviewer #1: Yes

Reviewer #2: Yes

5. Review Comments to the Author

Reviewer #1: The authors successfully investigated the relationship between late onset AD and DNA methylation levels in the top six Alzheimer’s disease (AD)-related genes in blood and examined its applicability to the diagnosis of AD. They also examined methylation differences at CpG island shores in the top six genes using Sanger sequencing, and one of two groups of 48 AD patients and 48 elderly controls was used for a test or replication analysis. They found that methylation levels in three out of the six genes, CR1, CLU, and PICALM, were lower in AD subjects. The combination of CLU methylation levels and the APOE genotype classified AD patients with AUC=0.84 and 0.80 in the test and replication analyses, respectively. These results indicated that the methylation differences at the CpG island shores of AD related genes in the onset of AD and suggested their diagnostic value.

I have several major comments.

1. In the Abstract, top six genes they examined should be shown.

2. In the Introduction line 63, the citations should be provided for the following sentence. “CpG islands are exempt from DNA methylation irrespective of gene expression, and the further away CpG dinucleotides are from CpG islands, the higher the chance of methylation.”. Is this well-known phenomenon?

3. In the Methods line 98, The reasons for selecting sanger sequencing, pyrosequencing, or by both methods should be clarified. Why did not authors select the consistent method?

4. In the Methods line 116, the authors used a program named CyGnusPlotter to identify CpG island shores in AD-associated genes. To my knowledge, this program seems not to be used commonly to identify the CpG iland shores. Please elaborate on the principles of the program and cite other papers using this program. Validity of this program is pivotal for this paper. I wonder if the selected CpG sites by this program (CpG shores in this paper) actually affect the gene expression.

5. In the Methods, the quality of pyrosequencing methylation data should be explained. Did all data pass the quality check? Could you show the typical peaks in suppl figures?

6. In statistical analysis, multiple comparisons should be corrected because they examined the mean methylation rates of 7 genes including TREM2.

7. In the Results line 150, the authors selected target regions in the CpG island shores of six known AD-related genes in the AlzGene database however this database is not appropriate for selecting target genes (This database included GWAS meta-analysis results published in 2013 by the International Genomics of Alzheimer’s Project (IGAP) consortium). To my knowledge, the latest top AD associated genes are shown in the GWAS paper below.

Kunkle et al. Genetic meta-analysis of diagnosed Alzheimer's disease identifies new risk loci and implicates Aβ, tau, immunity and lipid processing. Nat Genet. 2019 Mar;51(3):414-430.

8. In the Results line 185, the authors stated only positive results about PICALM, CR1, and CLU. However, the negative results about (APOE, BIN1 and ABCA7) also should be shown in the text and discussed. I think similar papers below should be discussed.

Mise A et al: TOMM40 and APOE Gene Expression and Cognitive Decline in Japanese Alzheimer's Disease Subjects. J Alzheimers Dis 60:1107-1117,2017.

Yamazaki K et al: Gene Expression and Methylation Analysis of ABCA7 in Patients with Alzheimer's Disease. J Alzheimers Dis 57:171-181,2017.

9. In the Results line 310, if the authors would like to show the impact of diagnostic values of methylation levels, the number of APOE4 alleles should not be included as covariates since the APOE epsilon4 allele is the strongest genetic risk factor for AD.

Reviewer #2: The article presents an interesting approach to the global methylation of genetic risk factors for Alzheimer's disease. I would like the title to be revised since the variations in SNPs and methylation tend to be different between ethnic groups, therefore the title should carry the target population "Japanese or asiactic population".

6. PLOS authors have the option to publish the peer review history of their article (what does this mean?). If published, this will include your full peer review and any attached files.

**Do you want your identity to be public for this peer review?** For information about this choice, including consent withdrawal, please see our Privacy Policy.

Reviewer #1: Yes: Jun-ichi Iga

Reviewer #2: Yes: David Salcedo-Tacuma

We would appreciate receiving your revised manuscript by October, 2020. To enhance the reproducibility of your results, we recommend that if applicable you deposit your laboratory protocols in protocols.io, where a protocol can be assigned its own identifier (DOI) such that it can be cited independently in the future. For instructions see: http://journals.plos.org/plosone/s/submission-guidelines#loc-laboratory-protocols

We look forward to receiving your revised manuscript.

Kind regards,

Stephen D. Ginsberg, Ph.D.

Section Editor

PLOS ONE

1. Please describe in your methods section how capacity to provide consent was determined for the participants in this study. Please also state whether your ethics committee or IRB approved this consent procedure. If you did not assess capacity to consent please briefly outline why this was not necessary in this case.

2.  Thank you for including your competing interests statement; "The authors have declared that no competing interests exist."

We note that one or more authors are affiliated with; Axcelead Drug Discovery Partners, Inc., Fujisawa, Kanagwa, Japan

---

## [Author Response · Author response to Decision Letter 0]

22 Aug 2020

Answers to Reviewer #1

1. As Reviewer #1 requested, we included the top six genes, CLU, CR1, PICALM, BIN1, APOE, and ABCA7, in the Abstract.

2. As Reviewer #1 requested, we cited appropriate references for the following sentence: “CpG islands are exempt from DNA methylation irrespective of gene expression, and the further away CpG dinucleotides are from a CpG island, the higher the chance of methylation.” in Introduction line 63. We consider it a well-known phenomenon as the references have been cited in many reports.

3. As Reviewer #1 requested, we described the reason for selecting Sanger sequencing, pyrosequencing, or by both methods in Materials and Methods line 104. As Sanger sequencing can quantify methylation levels of longer regions than pyrosequencing with in the same time, we selected it to identify the locations of CpG island shores in target genes. Sanger sequencing was again employed to find such CpG island shores where unidirectional changes in DNA methylation occur at consecutive CpGs. This was because DNA methylation changes are not at isolated CpGs but at consecutive CpGs in a gene accompanying biologically meaningful changes such as modification of chromatin and consequent alternation of gene activity. To confirm methylation changes identified by Sanger sequencing, we employed pyrosequencing because it is much higher throughput.

4. Contrary to Reviewer #1’s indication, “CyGnusPlotter” is not the program to predict possible locations of CpG island shores in a gene. It simply draws the positions of CpG dinucleotides along the gene of interest. We chose candidate CpG island shores in the plots by eye, judging from the density of the CpGs. They were not confirmed as CpG island shores until they were found to be slightly (10~20%) or moderately (~50%) methylated. This is described in the section of “Identification of CpG island shores” in the Materials and methods.

Reviewer #1 wonders if the methylation changes in CpG island shores affect gene expression. We examined the possibility in a group (32) of our subjects and presented the data in the new Results section “Correlation between gene expression and methylation levels” (line 234) and in S8 Fig. No significant expression changes were observed with the sample set.

5. As Reviewer #1 requested, we explained the quality of pyrosequencing data in Methods line 120. Not all data passed the quality check and we reanalyzed failed data, which are shown as red scores by PyroMark Q48 Autoprep Software. We added the figure that shows the typical peaks of pyrosequencing (Pyrogram) in S7 Fig.

6. As the Reviewer #1 pointed out, we should have considered multiple comparisons for the statistical analyses of DNA methylation differences in Fig 2. We set p=0.00625 (0.05 divided by 8) as the threshold for a significant difference for the analyses and revised the figure accordingly.

7. As Reviewer #1 pointed out, we agree that it was inappropriate to select the top six known AD-related genes from the database, AlzGene, because it relies on the GWAS meta-analysis published in 2013. Instead, as Reviewer #1 suggested, we cited the latest GWAS meta-analysis data, in which the six genes we selected were included in the top seven AD-related genes. The Abstract, Results line 166, and Discussion line 370 were revised.

8. As Reviewer #1 requested, we described the absence of DNA methylation difference in APOE, BIN1, and ABCA7 in Results line 204. We briefly discussed the negative results in Discussion line 374 and referred to the two papers that Reviewer #1 noted.

9. Reviewer #1 suggested that the number of APOE �4 alleles should not be included as covariates if we want to demonstrate the impact of diagnostic values of methylation levels. Unfortunately, the diagnostic values of methylation differences alone in any of the three genes, CLU, CR1, and PICALM, fell far short of diagnostic purposes, as we described in Results line 340 in the revised manuscript. For blood-based AD diagnosis, we do not stick to DNA methylation differences alone and adopt anything that can be utilized for better diagnosis. In this sense, APOE genotyping provides ideal information because it comes from blood DNA and has high diagnostic value; therefore, we would like to utilize it in combination with methylation differences.

An answer to Reviewer #2

1. As Reviewer #1 requested, we added “Japanese” in the title of our manuscript.

 Answers to the eight comments, [A1] ~ [A8], in the Word file of the manuscript:

[A1]: We added “Japanese” in the title as requested.

[A2]: We thank a reviewer for pointing out the mistake. We used peripheral blood leukocytes, not serum, and DNA was isolated. We corrected the mistake in Materials and methods line 89.

[A3]: We confirmed that the primers that were predicted to be useless in slico actually worked. The evidence is shown in S12 Fig.

[A4] & [A5]: we meant “slightly” and “moderately” for 10~20% and approximately 50%, respectively. To be clear, we also included the actual predicted percentages in parentheses in Results line 174 and in the Figure 1 legend line 195.

[A6]: We detected no significant difference at any single CpG site in the amplicons from the six genes, ABCA7, APOE, BIN1, CLU1, CR1, and PICALM, after Bonferroni correction, which was shown in S2 Table, and we described this in Discussion line 410. However, methylation differences with biological significance may be overlooked by Bonferroni correction for multiple testing. Using JASPAR 2020, we found that, among the CpG sites above, there are some with low p-values to which transcription factors can bind. We expect sufficient methods for multiple testing to be found ex post facto when changes in methylation affect transcription through such factors.

[A7]: As stated in Discussion line 380 in the revised manuscript, whether the methylation differences we found in AD blood are Japanese specific cannot be confirmed until they are examined in other ethnic populations. As the reviewer noted, a decrease in methylation at specific loci may be due to the decline in the DNA methyltransferase family, but to the best of our knowledge, no studies have revealed that it is a population-specific phenomenon or that any population-specific methylation differences occur in AD or any other diseases. We think it is one of the important issues to be addressed in our future studies.

[A8]: We appreciate this comment, which led us to a new finding. Stratification of APOE methylation by APOE genotype revealed a higher methylation level in �4 homozygotes, as shown in S9 Fig, suggesting the possibility that higher methylation at the APOE CpG island shore region and APOE4 synergistically affect the onset of AD. We described the data in Results line 394.

---

## [Decision Letter · Decision Letter 1]

2 Sep 2020

Lower DNA methylation levels in CpG island shores of CR1, CLU, and PICALM genes in the blood of Japanese Alzheimer’s disease patients

PONE-D-20-06667R1

Dear Dr. Shimoda,

We’re pleased to inform you that your manuscript has been judged scientifically suitable for publication and will be formally accepted for publication once it meets all outstanding technical requirements.

Kind regards,

Stephen D. Ginsberg, Ph.D.

Section Editor

PLOS ONE

**Comments to the Author**

1. If the authors have adequately addressed your comments raised in a previous round of review and you feel that this manuscript is now acceptable for publication, you may indicate that here to bypass the “Comments to the Author” section, enter your conflict of interest statement in the “Confidential to Editor” section, and submit your "Accept" recommendation.

Reviewer #1: All comments have been addressed

Reviewer #2: All comments have been addressed

2. Is the manuscript technically sound, and do the data support the conclusions?

Reviewer #1: Yes

Reviewer #2: Yes

3. Has the statistical analysis been performed appropriately and rigorously? 

Reviewer #1: Yes

Reviewer #2: Yes

4. Have the authors made all data underlying the findings in their manuscript fully available?

Reviewer #1: Yes

Reviewer #2: Yes

5. Is the manuscript presented in an intelligible fashion and written in standard English?

Reviewer #1: Yes

Reviewer #2: Yes

6. Review Comments to the Author

Reviewer #1: The authors successfully incorparated the reviewer's comments into the revised manuscript. I think this paper is suitable for publication in the current form.

Reviewer #2: After reading the manuscript and the reviews comprehensively, in my opinion the authors were able to satisfy my comments and the study is totally acceptable for publication with the corrections made. I have no further comment.

7. PLOS authors have the option to publish the peer review history of their article (what does this mean?). If published, this will include your full peer review and any attached files.

Reviewer #1: **Yes: **Junichi Iga

Reviewer #2: **Yes: **David Salcedo-Tacuma

---

## [Editor Report · Acceptance letter]

21 Sep 2020

PONE-D-20-06667R1 

Lower DNA methylation levels in CpG island shores of *CR1 *, *CLU*, and *PICALM* in the blood of Japanese Alzheimer’s disease patients 

Dear Dr. Shimoda:

I'm pleased to inform you that your manuscript has been deemed suitable for publication in PLOS ONE. Congratulations! Your manuscript is now with our production department. 

Kind regards, 

on behalf of

Dr. Stephen D. Ginsberg 

Section Editor

PLOS ONE